# Regret Analysis for Continuous Dueling Bandit

**Wataru Kumagai**
Center for Advanced Intelligence Project
RIKEN
1-4-1, Nihonbashi, Chuo, Tokyo 103-0027, Japan
`wataru.kumagai@riken.jp`

## Abstract

The dueling bandit is a learning framework wherein the feedback information in the learning process is restricted to a noisy comparison between a pair of actions. In this research, we address a dueling bandit problem based on a cost function over a continuous space. We propose a stochastic mirror descent algorithm and show that the algorithm achieves an $O(\sqrt{T \log T})$-regret bound under strong convexity and smoothness assumptions for the cost function. Subsequently, we clarify the equivalence between regret minimization in dueling bandit and convex optimization for the cost function. Moreover, when considering a lower bound in convex optimization, our algorithm is shown to achieve the optimal convergence rate in convex optimization and the optimal regret in dueling bandit except for a logarithmic factor.

## 1 Introduction

Information systems and computer algorithms often have many parameters which should be tuned. When cost or utility are explicitly given as numerical values or concrete functions, the system parameters can be appropriately determined depending on the the values or the functions. However, in a human-computer interaction system, it is difficult or impossible for users of the system to provide user preference as numerical values or concrete functions. *Dueling bandit* is introduced to model such situations in Yue and Joachims (2009) and enables us to appropriately tune the parameters based only on comparison results on two parameters by the users. In the learning process of a dueling bandit algorithm, the algorithm chooses a pair of parameters called actions (or arms) and receives only the corresponding comparison result. Since dueling bandit algorithms do not require an individual evaluation value for each action, they can be applied for wider areas that cannot be formulated using the conventional bandit approach.

When action cost (or user utility) implicitly exists, the comparison between two actions is modeled via a cost (or utility) function, which represents the degree of the cost (or utility), and a link function, which determines the noise in the comparison results. We refer to such a modeling method as cost-based (or utility-based) approach and employ it in this research. Yue and Joachims (2009) first introduced the utility-based approach as a model for a dueling bandit problem.

The cost-based dueling bandit relates to function optimization with noisy comparisons (Jamieson et al., 2012; Matsui et al., 2016) because in both frameworks an oracle compare two actions and the feedback from the oracle is represented by binary information. In particular, the same algorithm can be applied to both frameworks. However, as different performance measures are applied to the algorithms in function optimization and dueling bandit, it has not been demonstrated that an algorithm that works efficiently in one framework will also perform well in the other framework. This study clarifies relation between function optimization and dueling bandit thorough their regret analysis.

## 1.1 Problem Setup

In the learning process of the dueling bandit problem, a learner presents two points, called actions in a space $\mathcal{A}$, to an oracle and the oracle returns one-bit feedback to the learner based on which action wins (i.e., which action is more preferable for the oracle). Here, we denote by $a \succ a'$ the event that $a$ wins $a'$ and by $P(a \succ a')$ the probability that $a \succ a'$ happens. In other words, we assume that the feedback from the oracle follows the following two-valued random variable:

$$F(a, a') := \begin{cases} 1 & w.p. \quad P(a \succ a') \\ 0 & w.p. \quad 1 - P(a \succ a'), \end{cases} \tag{1}$$

where the probability $P(a \succ a')$ is determined by the oracle. We refer to this type of feedback as *noisy comparison feedback*. Unlike conventional bandit problems, the leaner has to make a decision that is based only on the noisy comparison feedback and cannot access the individual values of the cost (or utility) function. We further assume that each comparison between a pair of actions is independent of other comparisons.

The learner makes a sequence of decisions based on the noisy comparisons provided by the oracle. After receiving $F(a_t, a'_t)$ at time $t$, the learner chooses the next two action $(a_{t+1}, a'_{t+1})$. As a performance measure for an action $a$, we introduce the minimum win probability:

$$P^*(a) = \inf_{a' \in \mathcal{A}} P(a \succ a').$$

We next quantify the performance of the algorithm using the expected regret as follows:[1]

$$Reg_T^{DB} = \sup_{a \in \mathcal{A}} \mathbb{E}\left[\sum_{t=1}^{T} \{(P^*(a) - P^*(a_t)) + (P^*(a) - P^*(a'_t))\}\right]. \tag{2}$$

## 1.2 Modeling Assumption

In this section, we clarify some of the notations and assumptions. Let an action space $\mathcal{A} \subset \mathbb{R}^d$ be compact convex set with non-empty interior. We denote the Euclidean norm by $\|\cdot\|$.

**Assumption 1.** *There exist functions $f : \mathcal{A} \to \mathbb{R}$ and $\sigma : \mathbb{R} \to [0,1]$ such that the probability in noisy comparison feedback can be represented as follows:*

$$P(a \succ a') = \sigma(f(a') - f(a)). \tag{3}$$

In the following, we call $f$ in Assumption 1 a cost function and and $\sigma$ a link function. Here, the cost function and the link function are fixed for each query to the oracle. In this sense, our setting is different from online optimization where the objective function changes.

**Definition 1.** *(Strong Convexity) A function $f : \mathbb{R}^d \to \mathbb{R}$ is $\alpha$-strongly convex over the set $\mathcal{A} \subset \mathbb{R}^d$ if for all $x, y \in \mathcal{A}$ it holds that*

$$f(y) \geq f(x) + \nabla f(x)^\top (y - x) + \frac{\alpha}{2}\|y - x\|^2.$$

**Definition 2.** *(Smoothness) A function $f : \mathbb{R}^d \to \mathbb{R}$ is $\beta$-smooth over the set $\mathcal{A} \subset \mathbb{R}^d$ if for all $x, y \in \mathcal{A}$ it holds that*

$$f(y) \leq f(x) + \nabla f(x)^\top (y - x) + \frac{\beta}{2}\|y - x\|^2.$$

**Assumption 2.** *The cost function $f : \mathcal{A} \to \mathbb{R}$ is twice continuously differentiable, L-Lipschitz, $\alpha$-strongly convex and $\beta$-smooth with respect to the Euclidean norm.*

From Assumption 2, there exists a unique minimizer $a^*$ of the cost function $f$ since $f$ is strictly convex. We set $B := \sup_{a, a' \in \mathcal{A}} f(a') - f(a)$.

**Assumption 3.** *The link function $\sigma : \mathbb{R} \to [0,1]$ is three times differentiable and rotation-symmetric (i.e., $\sigma(-x) = 1 - \sigma(x)$). Its first derivative is positive and monotonically non-increasing on $[0, B]$.*

For examples, the standard logistic distribution function, the cumulative standard Gaussian distribution function and the linear function $\sigma(x) = (1 + x)/2$ can be taken to be link functions that satisfy Assumption 3. We note that link functions often behave like cumulative probability distribution functions. This is because the sign of the difference between two noisy function values can be regarded as the feedback (1) which satisfies Assumption 1, and then, the link function $\sigma$ coincides with the cumulative probability distribution function of the noise (see Section 2 of Jamieson et al. (2012) for more details). We will discuss the relation of noisy comparison feedback to noisy function values in Section 5.

## 1.3 Related Work and Our Contributions

Dueling bandit on the continuous action space relates with various optimization methods. We summarize related studies in the following.

**Dueling bandit problem:** Yue and Joachims (2009) formulated information retrieval systems as a dueling bandit problem. They reduced this to a problem of optimizing an "almost"-concave function and presented a stochastic gradient ascent algorithm based on one-point bandit feedback. Subsequently, they showed that their algorithm achieves an $O(T^{3/4})$-regret bound under the differentiability and the strict concavity for a utility function. Ailon et al. (2014) presented reduction methods from dueling bandit to the conventional bandit under the strong restriction that the link function is linear and showed that their algorithm achieves an $O(\sqrt{T \log^3 T})$-regret bound. We note that dueling bandit has a number of other formulations (Yue and Joachims, 2011; Yue et al., 2012; Busa-Fekete et al., 2013, 2014; Urvoy et al., 2013; Zoghi et al., 2014; Jamieson et al., 2015).

**Optimization with one-point bandit feedback:** In conventional bandit settings, various convex optimization methods have been studied. Flaxman et al. (2005) showed that the gradient of smoothed version of a convex function can be estimated from a one-point bandit feedback and proposed a stochastic gradient descent algorithm which achieves an $O(T^{3/4})$ regret bound under the Lipschitzness condition. Moreover, assuming the strong convexity and the smoothness for the convex function such as (2), Hazan and Levy (2014) proposed a stochastic mirror descent algorithm which achieves an $O(\sqrt{T \log T})$ regret bound and showed that the algorithm is near optimal because the upper bound matched the lower bound of $\Omega(\sqrt{T})$ derived by Shamir (2013) up to a logarithmic factor in bandit convex optimization.

**Optimization with two-point bandit feedback:** Dueling bandit algorithms require two actions at each round in common with two-point bandit optimization. In the context of online optimization, Agarwal et al. (2010) first considered convex optimization with two-point feedback. They proposed a gradient descent-based algorithm and showed that the algorithm achieves the regret bounds of under the Lipschitzness condition and $O(\log T)$ under the strong convexity condition. In stochastic convex optimization, Duchi et al. (2015) showed that a stochastic mirror descent algorithm achieves an $O(\sqrt{T})$ regret bound under the Lipschitzness (or the smoothness) condition and proved the upper bound to be optimal deriving a matching lower bound $\Omega(\sqrt{T})$. Moreover, in both of online and stochastic convex optimization, Shamir (2017) showed that a gradient descent-based algorithm achieves an $O(\sqrt{T})$ regret bound with optimal dependence on the dimension under the Lipschitzness condition. However, those two-point bandit algorithms strongly depend on the availability of the difference of function values and cannot be directly applied to the case of dueling bandit where the difference of function values is compressed to one bit in noisy comparison feedback.

**Optimization with noisy comparison feedback:** The cost-based dueling bandit relates to function optimization with noisy comparisons (Jamieson et al., 2012; Matsui et al., 2016) because in both frameworks, the feedback is represented by preference information. Jamieson et al. (2012) proposed a coordinate descent algorithm and proved that the convergence rate of the algorithm achieved an optimal order.[2] Matsui et al. (2016) proposed a Newton method-based algorithm and proved that its convergence rate was almost equivalent to that of Jamieson et al. (2012). They further showed that their algorithm could easily be parallelized and performed better numerically than the dueling bandit algorithm in Yue and Joachims (2009). However, since they considered only the unconstrained case in which $\mathcal{A} = \mathbb{R}^d$, it is not possible to apply their algorithm to the setting considered here, in which the action space is compact.

**Optimization with one-bit feedback:** The optimization method of the dueling bandit algorithm is based on one-bit feedback. In related work, Zhang et al. (2016) considered stochastic optimization under one-bit feedback. However, since their approach was restricted to the problem of linear optimization with feedback generated by the logit model, it cannot be applied to the problem addressed in the current study.

**Our contributions:** In this paper, we consider the cost-based dueling bandit under Assumptions 1-3. While the formulation is similar to that of Yue and Joachims (2009), Assumptions 2 and 3 are stronger than those used in that study. On the other hand, we impose the weaker assumption on the link function than that of Ailon et al. (2014). Yue and Joachims (2009) showed that a stochastic gradient descent algorithm can be applied to dueling bandit. Thus, it is naturally expected that a stochastic mirror descent algorithm, which achieves the (near) optimal order in convex optimization with one/two-point bandit feedback, can be applied to dueling bandit setting and achieves good performance. Following this intuition, we propose a mirror descent-based algorithm. Our key contributions can be summarized as follows:

- We propose a stochastic mirror descent algorithm with noisy comparison feedback.
- We provide an $O(\sqrt{T \log T})$-regret bound for our algorithm in dueling bandit.
- We clarify the relation between the cost-based dueling bandit and convex optimization in terms of their regrets and show that our algorithm can be applied to convex optimization.
- We show that the convergence rate of our algorithm is $O(\sqrt{\log T / T})$ in convex optimization.
- We derive a lower bound in convex optimization with noisy comparison feedback and show our algorithm to be near optimal in both dueling bandit and convex optimization.

## 2 Algorithm and Main Result

### 2.1 Stochastic Mirror Descent Algorithm

We first prepare the notion of a self-concordant function on which our algorithm is constructed (see e.g., Nesterov et al. (1994), Appendix F in Griva et al. (2009)).

**Definition 3.** *A function $\mathcal{R} : \mathrm{int}(\mathcal{A}) \to \mathbb{R}$ is considered self-concordant if the following two conditions hold:*

1. *$\mathcal{R}$ is three times continuously differentiable and convex, and approaches infinity along any sequence of points approaching the boundary of $\mathrm{int}(\mathcal{A})$.*

2. *For every $h \in \mathbb{R}^d$ and $x \in int(\mathcal{A})$, $|\nabla^3 \mathcal{R}(x)[h,h,h]| \le 2(h^\top \nabla^2 \mathcal{R}(x)h)^{\frac{3}{2}}$ holds, where $\nabla^3 \mathcal{R}(x)[h,h,h] := \frac{\partial^3 \mathcal{R}}{\partial t_1 \partial t_2 \partial t_3}(x + t_1 h + t_2 h + t_3 h)\big|_{t_1=t_2=t_3=0}$.*

*In addition to these two conditions, if $\mathcal{R}$ satisfies the following condition for a positive real number $\nu$, it is called a $\nu$-self-concordant function:*

3. *For every $h \in \mathbb{R}^d$ and $x \in int(\mathcal{A})$, $|\nabla \mathcal{R}(x)^\top h| \le \nu^{\frac{1}{2}} (h^\top \nabla^2 \mathcal{R}(x)h)^{\frac{1}{2}}$.*

In this paper, we assume the Hessian $\nabla^2 \mathcal{R}(a)$ of a $\nu$-self-concordant function to be full-rank over $\mathcal{A}$ and $\nabla \mathcal{R}(\mathrm{int}(\mathcal{A})) = \mathbb{R}^d$. Bubeck and Eldan (2014) showed that such a $\nu$-self-concordant function satisfying $\nu = (1 + o(1))d$ will always exist as long as the dimension $d$ is sufficiently large. We next propose Algorithm 1, which we call *NC-SMD*. This can be regarded as *stochastic mirror descent* with noisy comparison feedback.

We make three remarks on Algorithm 1. First, the function $\mathcal{R}_t$ is self-concordant though not $\nu$-self-concordant. The second remark is as follows. Let us denote the local norms by $\|a\|_w = \sqrt{a^\top \nabla^2 \mathcal{R}(w)a}$. Then, if $\mathcal{R}$ is a self-concordant function for $\mathcal{A}$, the Dikin Ellipsoid $\{a' \in \mathcal{A}| \|a' - a\|_a \le 1\}$ centered at $a$ is entirely contained in $\mathrm{int}(\mathcal{A})$ for any $a \in \mathrm{int}(\mathcal{A})$. Thus, $a'_t := a_t + \nabla^2 \mathcal{R}_t(a_t)^{-\frac{1}{2}} u_t$ in Algorithm 1 is contained in $\mathrm{int}(\mathcal{A})$ for any $a_t \in \mathrm{int}(\mathcal{A})$ and a unit vector $u_t$. This shows a comparison between actions $a_t$ and $a'_t$ to be feasible. Our third remark is as follows. Since the self-concordant function $\mathcal{R}_t$ at round $t$ depends on the past actions $\{a_i\}_{i=1}^t$, it may be thought that those past actions are stored during the learning process. However, note that only $\nabla \mathcal{R}_t$

---

**Algorithm 1** Noisy Comparison-based Stochastic Mirror Descent (NC-SMD)

---

**Input:** Learning rate $\eta$, $\nu$-self-concordant function $\mathcal{R}$, time horizon $T$, tuning parameters $\lambda, \mu$
**Initialize:** $a_1 = \operatorname{argmin}_{a \in \mathcal{A}} \mathcal{R}(a)$.
**for** $t = 1$ to $T$ **do**
     Update $\mathcal{R}_t(a) = \mathcal{R}(a) + \frac{\lambda\eta}{2} \sum_{i=1}^t \|a - a_i\|^2 + \mu\|a\|^2$
     Pick a unit vector $u_t$ uniformly at random
     Compare $a_t$ and $a'_t := a_t + \nabla^2 \mathcal{R}_t(a_t)^{-\frac{1}{2}} u_t$ and receive $F(a'_t, a_t)$
     Set $\hat{g}_t = F(a'_t, a_t) d \nabla^2 \mathcal{R}_t(a_t)^{\frac{1}{2}} u_t$
     Set $a_{t+1} = \nabla \mathcal{R}_t^{-1}(\nabla \mathcal{R}_t(a_t) - \eta \hat{g}_t)$
**end for**
**Output:** $a_{T+1}$

---

and $\nabla^2 \mathcal{R}_t$ are used in the algorithm; $\nabla \mathcal{R}_t$ depends only on $\sum_{i=1}^t a_i$ and $\nabla^2 \mathcal{R}_t$ does not depend on the past actions. Thus, only the sum of past actions must be stored, rather than all past actions.

## 2.2 Main Result: Regret Bound

From Assumption 2 and the compactness of $\mathcal{A}$, the diameter $R$ and $B := \sup_{a,a' \in \mathcal{A}} f(a') - f(a)$ are finite. From Assumption 3, there are exist positive constants $l_0, L_0, B_2$ and $L_2$ such that the first derivative $\sigma'$ of the link function is bounded as $l_0 \leq \sigma' \leq L_0$ on $[-B, B]$ and the second derivative $\sigma''$ is bounded above by $B_2$ and $L_2$-Lipschitz on $[-B, B]$. We use the constants below.

The following theorem shows that with appropriate parameters, NC-SMD (Algorithms 1) achieves an $O(\sqrt{T \log T})$-regret bound.

**Theorem 4.** *We set $C := \nu + \frac{B_2 L^2 + (L+1)L_0\beta}{2\lambda}$. When the tuning parameters satisfy $\lambda \leq l_0\alpha/2$, $\mu \geq \left(L_0^3 L_2/\lambda\right)^2$ and the total number $T$ of rounds satisfies $T \geq C \log T$. Algorithm 1 with a $\nu$-self-concordant function and the learning parameter $\eta = \frac{1}{2d}\sqrt{\frac{C \log T}{T}}$ achieves the following regret bound under Assumptions 1-3:*

$$Reg_T^{DB} \leq 4d\sqrt{CT \log T} + 2LL_0 R. \tag{4}$$

# 3 Regret Analysis

We prove Theorem 4 in this section. The proofs of lemmas in this section are provided in supplementary material.

## 3.1 Reduction to Locally-Convex Optimization

We first reduce the dueling bandit problem to a locally-convex optimization problem. We define $P_b(a) := \sigma(f(a) - f(b))$ for $a, b \in \mathcal{A}$ and $P_t(a) := P_{a_t}(a)$. For a cost function $f$ and a self-concordant function $\mathcal{R}$, we set $a^* := \operatorname{argmin}_{a \in \mathcal{A}} f(a)$, $a_1 := \operatorname{argmin}_{a \in \mathcal{A}} \mathcal{R}(a)$ and $a_T^* := \frac{1}{T}a_1 + (1 - \frac{1}{T})a^*$. The regret of dueling bandit is bounded as follows.

**Lemma 5.** *The regret of Algorithm 1 is bounded as follows:*

$$Reg_T^{DB} \leq 2\mathbb{E}\left[\sum_{t=1}^T (P_t(a_t) - P_t(a_T^*))\right] + \frac{LL_0\beta}{\lambda\eta} \log T + 2LL_0 R. \tag{5}$$

The following lemma shows that $P_b$ inherits the smoothness of $f$ globally.

**Lemma 6.** *The function $P_b$ is $(L_0\beta + B_2 L^2)$-smooth for an arbitrary $b \in \mathcal{A}$.*

Let $\mathbb{B}$ be the unit Euclidean ball, $\mathbb{B}(a, \delta)$ the ball centered at $a$ with radius $\delta$ and $\mathcal{L}(a, b)$ the line segment between $a$ and $b$. In addition, for $a, b \in \mathcal{A}$, let $\mathcal{A}_\delta(a, b) := \cup_{a' \in \mathcal{L}(a,b)} \mathbb{B}(a', \delta) \cap \mathcal{A}$. The following lemma guarantees the local strong convexity of $P_b$.

**Lemma 7.** *The function $P_b$ is $\frac{1}{2}l_0\alpha$-strongly convex on $\mathcal{A}_\delta(a^*, b)$ when $\delta \leq \frac{l_0\alpha}{2L_0^3 L_2}$.*

## 3.2 Gradient Estimation

We note that $a_t + \nabla^2 \mathcal{R}_t(a_t)^{-\frac{1}{2}} x$ for $x \in \mathbb{B}$ is included in $\mathcal{A}$ due to the properties of the Dikin ellipsoid. We introduce the smoothed version of $P_t$ over $\mathrm{int}(\mathcal{A})$:

$$\hat{P}_t(a) \quad := \quad \mathbb{E}_{x \in \mathbb{B}}[P_t(a + \nabla^2 \mathcal{R}_t(a_t)^{-\frac{1}{2}} x)] \tag{6}$$

$$= \quad \mathbb{E}_{x \in \mathbb{B}}[\sigma(f(a + \nabla^2 \mathcal{R}_t(a_t)^{-\frac{1}{2}} x) - f(a_t))]. \tag{7}$$

Next, we adopt the following estimator for the gradient of $\hat{P}_t$:

$$\hat{g}_t := F(a_t + \nabla^2 \mathcal{R}_t(a_t)^{-\frac{1}{2}} u_t, a_t) d \nabla^2 \mathcal{R}_t(a_t)^{\frac{1}{2}} u_t,$$

where $u_t$ is drawn from the unit surface $\mathbb{S}$ uniformly. We then derive the unbiasedness of $\hat{g}_t$ as follows.

**Lemma 8.**

$$\mathbb{E}[\hat{g}_t | a_t] = \nabla \hat{P}_t(a_t).$$

## 3.3 Regret Bound with Bregman Divergence

From Lemma 5, the regret analysis in dueling bandit is reduced to the minimization problem of the regret-like value of $P_t$. Since $P_t$ is globally smooth and locally strongly convex from Lemmas 6 and 7, we can employ convex-optimization methods. Moreover, since $\hat{g}_t$ is an unbiased estimator for the gradient of the smoothed version of $P_t$ from Lemma 8, it is expected that stochastic mirror descent (Algorithm 1) with $\hat{g}_t$ is effective to the minimization problem. In the following, making use of the property of stochastic mirror descent, we bound the regret-like value of $P_t$ by the Bregman divergence, and subsequently prove Theorem 4.

**Definition 9.** *Let $\mathcal{R}$ be a continuously differentiable strictly convex function on $\mathrm{int}(\mathcal{A})$. Then, the Bregman divergence associated with $\mathcal{R}$ is defined by*

$$D_{\mathcal{R}}(a, b) = \mathcal{R}(a) - \mathcal{R}(b) - \nabla \mathcal{R}(b)^\top (a - b).$$

**Lemma 10.** *When $\lambda \le l_0 \alpha / 2$ and $\mu \ge \left(L_0^3 L_2 / \lambda\right)^2$, the regret of Algorithm 1 is bounded for any $a \in \mathrm{int}(\mathcal{A})$ as follows:*

$$\mathbb{E}\left[\sum_{t=1}^{T}(P_t(a_t) - P_t(a))\right]$$

$$\le \frac{1}{\eta}\left(\mathcal{R}(a) - \mathcal{R}(a_1) + \mathbb{E}\left[\sum_{t=1}^{T} D_{\mathcal{R}_t^*}(\nabla \mathcal{R}(a_t) - \eta \hat{g}_t, \nabla \mathcal{R}(a_t))\right]\right) + \frac{L_0 \beta + B_2 L^2}{\lambda \eta} \log T, \tag{8}$$

*where $\mathcal{R}_t^*(a) := \sup_{x \in \mathbb{R}^d} \langle x, a \rangle - \mathcal{R}_t(a)$ is the Fenchel dual of $\mathcal{R}_t$.*

The Bregman divergence in Lemma 10 is bounded as follows.

**Lemma 11.** *When $\eta \le \frac{1}{2d}$, the sequence $a_t$ output by Algorithm 1 satisfies*

$$D_{\mathcal{R}_t^*}(\nabla \mathcal{R}_t(a_t) - \eta \hat{g}_t, \nabla \mathcal{R}_t(a_t)) \le 4d^2 \eta^2. \tag{9}$$

**[Proof of Theorem 4]** From Lemma 4 of Hazan and Levy (2014), the $\nu$-self-concordant function $\mathcal{R}$ satisfies

$$\mathcal{R}(a_T^*) - \mathcal{R}(a_1) \le \nu \log \frac{1}{1 - \pi_{a_1}(a_T^*)},$$

where $\pi_a(a') := \inf\{r \ge 0 | a + r^{-1}(a' - a)\}$ is the Minkowsky function. Since $\pi_{a_1}(a_T^*) \le 1 - T^{-1}$ from the definition of $a_T^*$, we obtain

$$\mathcal{R}(a_T^*) - \mathcal{R}(a_1) \le \nu \log T.$$

Note that the condition $\eta \le \frac{1}{2d}$ in Lemma 11 is satisfied due to $T \ge C \log T$. Combining Lemmas 5, 10 and 11, we have

$$Reg_T^{DB} \quad \le \quad \frac{2}{\eta}\left(\nu \log T + 4d^2 \eta^2 T\right) + \frac{L_0 \beta + D_{\sigma''} L^2}{\lambda \eta} \log T + \frac{L L_0 \beta}{l_0 \alpha \eta} + 2 L L_0 R$$

$$\le \quad \left(2\nu + \frac{B_2 L^2 + (L+1) L_0 \beta}{\lambda}\right) \frac{\log T}{\eta} + 8d^2 T \eta + 2 L L_0 R.$$

Thus, when $\eta$ is defined in Theorem 4, the regret bound (4) is obtained. ∎

## 4   Convergence Rate in Convex Optimization

In the previous sections, we considered the minimization problem for the regret of dueling bandit. In this section, as an application of the approach, we show that the averaged action of NC-SMD (Algorithm 1) minimize the cost function $f$ in (3).

To derive the convergence rate of our algorithm, we introduce the regret of function optimization and establish a connection between the regrets of dueling bandit and function optimization. In convex optimization with noisy comparison feedback, the learner chooses a pair $(a_t, a'_t)$ of actions in the learning process and suffers a loss $f(a_t) + f(a'_t)$. Then, the regret of the algorithms in function optimization is defined as follows:

$$Reg_T^{FO} \quad := \quad \mathbb{E}\left[\sum_{t=1}^T (f(a_t) - f(a^*)) + (f(a'_t) - f(a^*))\right], \tag{10}$$

where $a^* = \operatorname{argmin}_{a \in \mathcal{A}} f$.

Recalling that the positive constants $l_0$ and $L_0$ satisfy $l_0 \leq \sigma' \leq L_0$ on $[-B, B]$, where $B := \sup_{a, a' \in \mathcal{A}} f(a') - f(a)$, the regrets of function optimization (10) and dueling bandit (2) are related as follows:

**Lemma 12.**

$$\frac{Reg_T^{DB}}{L_0} \leq Reg_T^{FO} \leq \frac{Reg_T^{DB}}{l_0}. \tag{11}$$

Theorem 4 and Lemma 12 give an $O(\sqrt{T \log T})$-upper bound of the regret of function optimization in Algorithm 1 under the same conditions as Theorem 4. Given the convexity of $f$, the average of the chosen actions of any dueling bandit algorithm $\bar{a}_T := \frac{1}{2T} \sum_{t=1}^T (a_t + a'_t)$ satisfies

$$\mathbb{E}[f(\bar{a}_T) - f(a^*)] \leq \frac{Reg_T^{FO}}{2T}. \tag{12}$$

Thus, if an optimization algorithm has a sub-linear regret bound, the above online-to-batch conversion guarantees convergence to the optimal point.

**Theorem 13.** *Under Assumptions 1-3, the averaged action $\bar{a}_T$ satisfies the following when $T \geq C \log T$:*

$$\mathbb{E}[f(\bar{a}_T) - f(a^*)] \leq \frac{1}{l_0}\left(2d\sqrt{\frac{\nu \log T + C}{T}} + \frac{LL_0 R}{T}\right),$$

*where $C$ is the constant defined in Theorem 4.*

Theorem 13 shows the convergence rate of NC-SMD (Algorithm 1) to be $O(d\sqrt{\log T / T})$.

## 5   Lower Bound

We next derive a lower bound in convex optimization with noisy comparison feedback. To do so, we employ a lower bound of convex optimization with *noisy function feedback*. In a setting where the function feedback is noisy, we query a point $a \in \mathcal{A}$ and obtain a noisy function value $f(a) + \xi$, where $\xi$ is a zero-mean random variable with a finite second moment and independent for each query. [3]

**Theorem 14.** *Assume that the action space $\mathcal{A}$ is the $d$-dimensional unit Euclidean ball $\mathbb{B}_d$ and that the link function $\sigma_\mathcal{G}$ is the cumulative distribution function of the zero-mean Gaussian random variable with variance 2. Let the number of rounds $T$ be fixed. Then, for any algorithm with noisy comparison feedback, there exists a function $f$ over $\mathbb{B}_d$ which is twice continuously differentiable, 0.5-strongly convex and 3.5-smooth such that the output $a_T$ of the algorithm satisfies*

$$\mathbb{E}[f(a_T) - f(a^*)] \geq 0.004 \min\left\{1, \frac{d}{\sqrt{2T}}\right\}. \tag{13}$$

**[Proof]** The probability distribution of noisy comparison feedback $F(a, a')$ with the link function $\sigma_{\mathcal{G}}$ can be realized by noisy function feedback with the standard Gaussian noise as follows. Two noisy function values $f(a) + \xi$ and $f(a') + \xi'$ can be obtained using the noisy function feedback twice, where $\xi$ and $\xi'$ are independent standard Gaussian random variables. Then, the probability distribution of the following random variable coincide with that of $F(a, a')$ for arbitrary $a, a' \in \mathcal{A}$:

$$\text{sign}(f(a) + \xi - (f(a') + \xi')) = \text{sign}(f(a) - f(a') + (\xi - \xi')). \tag{14}$$

Here, note that $\xi - \xi'$ is the zero-mean Gaussian random variable with variance 2. Thus, single noisy comparison feedback with the link function $\sigma_{\mathcal{G}}$ for any actions can be obtained by using noisy function feedback with standard Gaussian noise twice. This means that if any algorithm with $2T$-times noisy function feedback is unable to achieve a certain performance, any algorithm with $T$-times noisy comarison feedback is similarly unable to achieve that performance. Thus, to derive Theorem 14, it is sufficient to show a lower bound of convergence rate with noisy function feedback. The following lower bound is derived by Theorem 7 of Shamir (2013).

**Theorem 15.** *(Shamir, 2013) Let the number of rounds $T$ be fixed. Suppose that the noise $\xi$ at each round is a standard Gaussian random variable. Then, for any algorithm with noisy function feedback, there exists a function $f$ over $\mathbb{B}_d$ which is twice continuously differentiable, $0.5$-strongly convex and $3.5$-smooth such that the output $a_T$ satisfies*

$$\mathbb{E}[f(a_T) - f(a^*)] \geq 0.004 \min\left\{1, \frac{d}{\sqrt{T}}\right\}.$$

By the above discussion and from Theorem 15, we obtain Theorem 14. ∎

Combining Theorem 13 and Theorem 14, the convergence rate of NC-SMD (Algorithm 1) is near optimal with respect to the number of rounds $T$. In addition, when the parameter $\nu$ of the self-concordant function is of constant order with respect to the dimension $d$ of the space $\mathcal{A}$, the convergence rate of NC-SMD is optimal with respect to $d$. However, it should be noted that the parameter $\nu$ of a self-concordant function is often of the order of $\Theta(d)$ for compact convex sets including the simplex and the hypercube.

As a consequece of Lemma 12, (12), and Theorems 4 and 14, the optimal regrets of dueling bandit and function optimization are of the order $\sqrt{T}$ except for the logarithmic factor and NC-SMD achieves the order. To the best of our knowledge, this is the first algorithm with the optimal order in the continuous dueling bandit setting with the non-linear link function.

Finally, we provide an interesting observation on convex optimization. When noisy function feedback is available, the optimal regret of function optimization is of the order $\Theta(\sqrt{T})$ under strong convexity and smoothness conditions (Shamir, 2013). However, even when noisy function feedback is "compressed" into one-bit information as in (14), our results show that NC-MSD (Algorithm 1) achieves almost the same order $O(\sqrt{T \log T})$ about the regret of function optimization as long as the cumulative probability distribution of the noise satisfies Assumption 3.[4]

# 6 Conclusion

We considered a dueling bandit problem over a continuous action space and proposed a stochastic mirror descent. By introducing Assumptions 1-3, we proved that our algorithm achieves an $O(\sqrt{T \log T})$-regret bound. We further considered convex optimization under noisy comparison feedback and showed that the regrets of dueling bandit and function optimization are essentially equivalent. Using the connection between the two regrets, it was shown that our algorithm achieves a convergence rate of $O(\sqrt{\log T / T})$ in the framework of function optimization with noisy comparison feedback. Moreover, we derived a lower bound of the convergence rate in convex optimization and showed that our algorithm achieves near optimal performance in dueling bandit and convex optimization. Some open questions still remain. While we have only dealt with bounds which hold in expectation, the derivation of the high-probability bound is a problem that has not been solved. While the analysis of our algorithm relies on strong convexity and smoothness, a regret bound without these conditions is also important.

## Acknowledgment

We would like to thank Professor Takafumi Kanamori for helpful comments. This work was supported by JSPS KAKENHI Grant Number 17K12653.

## Footnotes

[1] Although the regret in (2) appears superficially different from that in Yue and Joachims (2009), two regrets can be shown to coincide with each other under Assumptions 1-3 in Subsection 1.2.

[2]The optimal order changes depending on the model parameter $\kappa \geq 1$ of the pairwise comparison oracle in Jamieson et al. (2012).

[3] In general, the noise $\xi$ can depend on the action $a$. See e.g. Shamir (2013) for more details.

[4] Jamieson et al. (2012) provided a similar observation. However, their upper bound of the regret was derived only when the action space is the whole of Euclidean space (i.e., $\mathcal{A} = \mathbb{R}^d$) and the assumption for noisy comparison feedback is different from ours (Assumption 1).

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
