[Supplementary Material]

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

# Appendix for **Regret Analysis for Continuous Dueling Bandit**

## A Appendix: Technical Proofs

[**Proof of Lemma 5**] From direct calculation,

$$
\begin{aligned}
Reg_T^{DB} &= \mathbb{E}\left[\sum_{t=1}^{T}\{2\sigma(f(a_t) - f(a_t)) - \sigma(f(a^*) - f(a_t)) - \sigma(f(a^*) - f(a_t + \nabla^2 \mathcal{R}_t(a_t)^{-\frac{1}{2}} u_t))\}\right] \\
&= 2\mathbb{E}\left[\sum_{t=1}^{T}\{\sigma(f(a_t) - f(a_t)) - \sigma(f(a^*) - f(a_t))\}\right] \\
&\quad + \mathbb{E}\left[\sum_{t=1}^{T}\{\sigma(f(a^*) - f(a_t)) - \sigma(f(a^*) - f(a_t + \nabla^2 \mathcal{R}_t(a_t)^{-\frac{1}{2}} u_t))\}\right]. \\
&= 2\mathbb{E}\left[\sum_{t=1}^{T}(P_t(a_t) - P_t(a^*))\right] \\
&\quad + \mathbb{E}\left[\sum_{t=1}^{T}\{\sigma(f(a^*) - f(a_t)) - \sigma(f(a^*) - f(a_t + \nabla^2 \mathcal{R}_t(a_t)^{-\frac{1}{2}} u_t))\}\right].
\end{aligned}
$$

Here, we note that $f(a^*) - f(a) \leq 0$ for any $a \in \mathcal{A}$ due to the definition of $a^*$ and that $\sigma$ is convex on $(-\infty, 0)$ because the link function is rotation symmetric and its derivative is monotonically non-increasing on positive real numbers from Assumption 3. Thus, Jensen's inequality derives

$$
\begin{aligned}
&\mathbb{E}\left[\sigma(f(a^*) - f(a_t + \nabla^2 \mathcal{R}_t(a_t)^{-\frac{1}{2}} u_t))|a_t\right] \\
&\geq \sigma(\mathbb{E}[f(a^*) - f(a_t + \nabla^2 \mathcal{R}_t(a_t)^{-\frac{1}{2}} u_t)|a_t]) \\
&= \sigma(f(a^*) - \hat{f}(a_t)).
\end{aligned}
$$

In addition, $f(a_t) \leq \hat{f}(a_t)$ holds due to the convexity of $f$. As $\sigma$ is monotonically non-decreasing from Assumption 3, we have

$$
\begin{aligned}
&\mathbb{E}\left[\sum_{t=1}^{T}\{\sigma(f(a^*) - f(a_t)) - \sigma(f(a^*) - f(a_t + \nabla^2 \mathcal{R}_t(a_t)^{-\frac{1}{2}} u_t))\}\right] \\
&= \mathbb{E}\left[\mathbb{E}\left[\sum_{t=1}^{T}\{\sigma(f(a^*) - f(a_t)) - \sigma(f(a^*) - f(a_t + \nabla^2 \mathcal{R}_t(a_t)^{-\frac{1}{2}} u_t))\}|a_t\right]\right] \\
&\leq \mathbb{E}\left[\sum_{t=1}^{T}\{\sigma(f(a^*) - f(a_t)) - \sigma(f(a^*) - \hat{f}(a_t))\}\right] \\
&\leq LL_0\mathbb{E}\left[\sum_{t=1}^{T}(\hat{f}(a_t) - f(a_t))\right] \\
&\leq \frac{LL_0\beta}{2}\mathbb{E}\left[\sum_{t=1}^{T}\mathbb{E}_{x\in\mathbb{B}}[\|\nabla^2 \mathcal{R}_t(a_t)^{-\frac{1}{2}} x\|^2]\right] \\
&\leq \frac{LL_0\beta}{2}\mathbb{E}\left[\sum_{t=1}^{T}\frac{1}{\lambda\eta t}\right] \\
&\leq \frac{LL_0\beta}{\lambda\eta}\log T,
\end{aligned}
$$

where we used the property of the $\beta$-smoothness of $f$ in the third inequality. Thus, we obtain (5)

$$
Reg_T^{DB} \leq 2\mathbb{E}\left[\sum_{t=1}^{T}(P_t(a_t) - P_t(a^*))\right] + \frac{LL_0\beta}{\lambda\eta}\log T.
$$

Here, we have

$$\mathbb{E}\left[\sum_{t=1}^{T}(P_t(a_t) - P_t(a^*))\right] = \mathbb{E}\left[\sum_{t=1}^{T}(P_t(a_t) - P_t(a_T^*))\right] + \mathbb{E}\left[\sum_{t=1}^{T}(P_t(a_T^*) - P_t(a^*))\right].$$

From the definition of $a_T^*$, we have

$$P_t(a_T^*) - P_t(a^*) \leq LL_0\|a_T^* - a^*\| \leq \frac{LL_0 R}{T},$$

where $R$ is the diameter of $\mathcal{A}$. Thus (5) is obtained. ∎

[**Proof of Lemma 6**] From direct calculation, we obtain that

$$\nabla P_b(a) = \sigma'(f(a) - f(b))\nabla f(a), \tag{15}$$
$$\nabla^2 P_b(a) = \sigma'(f(a) - f(b))\nabla^2 f(a) + \sigma''(f(a) - f(b))\nabla f(a)\nabla f(a)^\top. \tag{16}$$

Then, it is sufficient to give upper bounds on the first and second terms in (16) in the sense of matrix inequalities as

$$\sigma'(f(a) - f(b))\nabla^2 f(a) \leq L_0\beta I, \tag{17}$$
$$\sigma''(f(a) - f(b))\nabla f(a)\nabla f(a)^\top \leq B_2 L^2 I, \tag{18}$$

where $I$ is the $d \times d$ identity matrix. The inequality (17) follows from the $L_0$-Lipschitzness of $\sigma$ and the $\beta$-smoothness of $f$. The inequality (18) follows from the $B_2$-boundedness of $\sigma''$ and the $L$-Lipschitzness of $f$. ∎

[**Proof of Lemma 7**] We first show that $P_b$ is $l_0\alpha$-strongly convex on $\mathcal{L}(a^*, b)$. Since (16) holds for any $a \in \mathcal{L}(a^*, b)$, it is sufficient to give lower bounds on the first and second terms in (16) for $a$ in $\mathcal{A}_t$ in the sense of matrix inequalities. In the following, we show

$$\sigma'(f(a) - f(b))\nabla^2 f(a) \geq l_0\alpha I, \tag{19}$$
$$\sigma''(f(a) - f(b))\nabla f(a)\nabla f(a)^\top \geq 0. \tag{20}$$

Since $l_0 \leq \sigma'$ and $f$ is $\alpha$-strongly convex, we obtian (19). Next, we show (20). Since $\sigma'$ is monotonically non-decreasing on $[-B, 0]$, $\sigma''(y)$ is negative only if $y$ is positive. Note that $f(a) - f(b) \leq 0$ for any $a \in \mathcal{L}(a^*, b)$ since $-f(b) + f(a^*) \leq 0$ and $f$ is convex. Thus, we have (20).

Next, we show that $P_b$ is $\frac{1}{2}l_0\alpha$-strongly convex on $\mathcal{A}_\delta(a^*, b)$ when $\delta \leq \frac{l_0\alpha}{4L_0^3 L_2}$. For an arbitrary $\tilde{a} \in \mathcal{A}_\delta(a^*, b)$, there exists $a \in \mathcal{L}(a^*, b)$ and $y \in \mathbb{B}(0, \delta)$ such that $\tilde{a} = a + y$ by the definition of $\mathcal{A}_\delta(a^*, b)$. Since (16) holds, it is sufficient to give lower bounds on the first and second terms in (16) for $a$ in $\mathcal{A}_t$ in the sense of matrix inequalities. Since (19) holds all we have to do is to show

$$\sigma''(f(a + y) - f(b))\nabla f(a + y)\nabla f(a + y)^\top \geq -\frac{1}{2}l_0\alpha I. \tag{21}$$

Since $\sigma'$ is monotonically non-decreasing on $[-B, 0]$, $\sigma''(z)$ is negative only if $z$ is positive. Note that $f(a) - f(b) \leq 0$ for any $a \in \mathcal{L}(a^*, b)$ since $f(a^*) - f(b) \leq 0$ and $f$ is convex. Thus, we have

$$\sigma''(f(a + y) - f(b))\nabla f(a + y)\nabla f(a + y)^\top$$
$$\geq \begin{cases} \sigma''(f(a + y) - f(b))L^2 I & if \quad f(a + y) - f(b) > 0 \\ 0 & if \quad f(a + y) - f(b) \leq 0. \end{cases} \tag{22}$$

When $f(a + y) - f(b) > 0$ and $a \in \mathcal{L}(b, a^*)$, the following holds by the $L$-Lipshitzness of $f$:

$$\begin{aligned} f(a + y) - f(b) &= f(a) - f(b) - f(a) + f(a + y) \\ &\leq -f(a) + f(a + y) \\ &\leq L\delta. \end{aligned}$$

where we used again $f(a) - f(b) \leq 0$ for any $a \in \mathcal{L}(a^*, b)$. Thus

$$\begin{aligned} \sigma''(f(a + y) - f(b)) &= \sigma''(f(a + y) - f(b)) - \sigma''(0) \\ &\geq -L_2(f(a + y) - f(b)) \\ &\geq -LL_2\delta. \end{aligned} \tag{23}$$

Combining (22) and (23), we have

$$\sigma''(f(a + y) - f(b))\nabla f(a + y)\nabla f(a + y)^\top \geq -L^3 L_2\delta I. \tag{24}$$

Thus, when $\delta \leq \frac{l_0\alpha}{2L_0^3 L_2}$, we obtain (21). ∎

[**Proof of Lemma 8**] We have

$$
\begin{aligned}
\mathbb{E}[\hat{g}_t|a_t] &= \mathbb{E}_{u_t}[\mathbb{E}[\hat{g}_t|a_t, u_t]] \\
&= \mathbb{E}_{u_t}[d\mathbb{E}[P_t(a_t + \nabla^2 \mathcal{R}_t(a_t)^{-\frac{1}{2}} u_t)\nabla^2 \mathcal{R}_t(a_t)^{\frac{1}{2}} u_t|a_t, u_t]] \\
&= d\mathbb{E}[P_t(a_t + \nabla^2 \mathcal{R}_t(a_t)^{-\frac{1}{2}} u_t)\nabla^2 \mathcal{R}_t(a_t)^{\frac{1}{2}} u_t|a_t] \\
&= \nabla \mathbb{E}_{x\in\mathbb{B}}[P_t(a_t + \nabla^2 \mathcal{R}_t(a_t)^{-\frac{1}{2}} x)|a_t] \qquad (25) \\
&= \nabla \hat{P}_t(a_t),
\end{aligned}
$$

where we used Stokes' theorem in (25). ∎

[**Proof of Lemma 10**] We can divide the left hand side of (8) into three parts:

$$
\begin{aligned}
&\mathbb{E}\left[\sum_{t=1}^{T}(P_t(a_t) - P_t(a))\right] \\
=\ &\mathbb{E}\left[\sum_{t=1}^{T}(\hat{P}_t(a_t) - \hat{P}_t(a))\right] + \mathbb{E}\left[\sum_{t=1}^{T}(P_t(a_t) - \hat{P}_t(a_t))\right] + \mathbb{E}\left[\sum_{t=1}^{T}(\hat{P}_t(a) - P_t(a))\right].
\end{aligned}
$$

Here, we bound the above three terms, respectively. First, let us recall that $P_t$ is strongly convex on $\mathbb{B}(a_t, \frac{l_0\alpha}{2L_0^3 L_2}) \cap \mathcal{A}$ due to Lemma 7. By the definition of $\mathcal{R}_t$ and the conditions for $\lambda$ and $\mu$, it holds that $a_t + \nabla^2 \mathcal{R}_t(a_t)^{-\frac{1}{2}} x \in \mathbb{B}(a_t, \frac{l_0\alpha}{2L_0^3 L_2}) \cap \mathcal{A}$ for any $x \in \mathbb{B}$. Thus, from the local convexity of $P_t$ in Lemma 7 and Jensen's inequality, $P_t(a_t) - \hat{P}_t(a_t) \leq 0$ holds. Next, from the smoothness of $P_t$ in Lemma 6, we have

$$
\hat{P}_t(a) - P_t(a) \leq \frac{L_0\beta + B_2 L^2}{2}\|\nabla^2 \mathcal{R}_t(a_t)^{-\frac{1}{2}} u\|^2 \leq \frac{L_0\beta + B_2 L^2}{2\lambda\eta t},
$$

where the first inequality follows from, for example, Lemma 7 of Hazan and Levy (2014) and the second inequality follows from the definition of $\mathcal{R}_t$. Hence, we obtain

$$
\mathbb{E}\left[\sum_{t=1}^{T}(\hat{P}_t(a) - P_t(a))\right] \leq \frac{L_0\beta + B_2 L^2}{\lambda\eta}\log T.
$$

Finally, we bound the first term of the upper bound of the regret. We have the following inequalities:

$$
\begin{aligned}
&\mathbb{E}[\hat{P}_t(a_t) - \hat{P}_t(a)] \\
\leq\ &\mathbb{E}\left[\nabla \hat{P}_t(a_t)^\top(a_t - a) - \frac{l_0\alpha}{4}\|a_t - a\|^2\right] \\
=\ &\mathbb{E}\left[\hat{g}_t^\top(a_t - a) - \frac{l_0\alpha}{4}\|a_t - a\|^2\right] \\
=\ &\eta^{-1}\mathbb{E}\left[(\nabla \mathcal{R}_t(a_{t+1}) - \nabla \mathcal{R}_t(a_t))^\top(a - a_t) - \frac{l_0\alpha\eta}{4}\|a_t - a\|^2\right] \\
=\ &\eta^{-1}\mathbb{E}\left[D_{\mathcal{R}_t}(a, a_t) + D_{\mathcal{R}_t}(a_t, a_{t+1}) - D_{\mathcal{R}_t}(a, a_{t+1}) - \frac{l_0\alpha\eta}{4}\|a_t - a\|^2\right],
\end{aligned}
$$

where the first inequality follows from the local convexity of $\hat{P}_t$, the first equality is derived by Lemma 8 and the second equality holds due to the definition of $a_{t+1}$. Summing up both sides,

$$
\mathbb{E}\left[\sum_{t=1}^{T}(\hat{P}_t(a_t) - \hat{P}_t(a))\right]
$$

$$
\leq \quad \eta^{-1}\mathbb{E}\left[D_{\mathcal{R}_1}(a, a_1) - D_{\mathcal{R}_T}(a, a_{T+1}) + \sum_{t=1}^{T} D_{\mathcal{R}_t}(a_t, a_{t+1})\right]
$$

$$
+ \eta^{-1}\mathbb{E}\left[\sum_{t=2}^{T}\left(D_{\mathcal{R}_t}(a, a_t) - D_{\mathcal{R}_{t-1}}(a, a_t) - \frac{l_0\alpha\eta}{4}\|a_t - a\|^2\right)\right]
$$

$$
\leq \quad \eta^{-1}\mathbb{E}\left[D_{\mathcal{R}_1}(a, a_1) - D_{\mathcal{R}_T}(a, a_T + 1) + \sum_{t=1}^{T} D_{\mathcal{R}_t}(a_t, a_{t+1})\right]
$$

$$
+ \eta^{-1}\mathbb{E}\left[\sum_{t=2}^{T}\left(D_{\mathcal{R}_t}(a, a_t) - D_{\mathcal{R}_{t-1}}(a, a_t) - \frac{\lambda\eta}{2}\|a_t - a\|^2\right)\right]
$$

$$
= \quad \eta^{-1}\mathbb{E}\left[D_{\mathcal{R}_1}(a, a_1) - D_{\mathcal{R}_T}(a, a_T + 1) + \sum_{t=1}^{T} D_{\mathcal{R}_t}(a_t, a_{t+1})\right]
$$

$$
\leq \quad \eta^{-1}\mathbb{E}\left[D_{\mathcal{R}_1}(a, a_1) + \sum_{t=1}^{T} D_{\mathcal{R}_t}(a_t, a_{t+1})\right]
$$

$$
= \quad \eta^{-1}\left(\mathcal{R}(a) - \mathcal{R}(a_1) + \mathbb{E}\left[\sum_{t=1}^{T} D_{\mathcal{R}_t^*}(\nabla\mathcal{R}_t(a_t) - \eta\hat{g}_t, \nabla\mathcal{R}_t(a_t))\right]\right)
$$

where we used the positivity of the Bregman divergence in the third inequality and $\nabla\mathcal{R}(a_1) = 0$ because $a_1 = \arg\min\mathcal{R}$ in the last equation. Combining the above discussion, we obtain (8). ∎

[**Proof of Lemma 11**] Taylar's theorem guarantees the existence of $\delta_t \in (0, 1)$ such that

$$
D_{\mathcal{R}_{t^*}}(\nabla\mathcal{R}_t(a_t) - \eta\hat{g}_t, \nabla\mathcal{R}_t(a_t)) = \eta^2\hat{g}_t^\top\nabla^2\mathcal{R}_t^*(\nabla\mathcal{R}_t(a_t) - \delta_t\eta\hat{g}_t)\hat{g}_t. \tag{26}
$$

Then using the self-concordant property of $\mathcal{R}_t^*$ (see e.g. (F.2) of Griva et al. (2009)),

$$
\hat{g}_t^\top\nabla^2\mathcal{R}_t^*(\nabla\mathcal{R}_t(a_t) - \delta_t\eta\hat{g}_t)\hat{g}_t \leq \left(\frac{\|\hat{g}_t\|_{\nabla\mathcal{R}_t(a_t)}^*}{1 - \delta_t\eta\|\hat{g}_t\|_{\nabla\mathcal{R}_t(a_t)}^*}\right)^2. \tag{27}
$$

where $\|x\|_y^* = \sqrt{x^\top\nabla^2\mathcal{R}_t^*(y)x}$. Here, we note that

$$
\begin{aligned}
\|\hat{g}_t\|_{\nabla\mathcal{R}_t(x_t)}^* &= \sqrt{\hat{g}_t^\top\nabla^2\mathcal{R}_t^*(\nabla\mathcal{R}_t(x_t))\hat{g}_t} \\
&= \sqrt{\hat{g}_t^\top\nabla^2\mathcal{R}_t(x_t)^{-1}\hat{g}_t} \\
&\leq d\sqrt{u_t^\top\nabla^2\mathcal{R}_t(x_t)^{\frac{1}{2}}\nabla^2\mathcal{R}_t(x_t)^{-1}\nabla^2\mathcal{R}_t(x_t)^{\frac{1}{2}}u_t} \\
&= d
\end{aligned} \tag{28}
$$

and thus, $\delta_t\eta\|\hat{g}_t\|_{\nabla\mathcal{R}(a_t)}^* < \frac{1}{2}$ when $\eta \leq \frac{1}{2d}$. Consequently, (26), (27) and (28) derives (9). ∎

[**Proof of Lemma 12**] We show the second inequality of (11). From the definition, Since $l_0 \leq \alpha_t'$ and $\sigma(0) = \frac{1}{2}$ from Assumption 3, we have

$$
\begin{aligned}
Reg_T^{DB} &= \sup_{a\in\mathcal{A}}\mathbb{E}\left[\sum_{t=1}^{T}\left(\{\sigma(f(a_t) - f(a)) - \sigma(0)\} + \{\sigma(f(a_t') - f(a)) - \sigma(0)\}\right)\right] \\
&\geq l_0\sup_{a\in\mathcal{A}}\mathbb{E}\left[\sum_{t=1}^{T}\left(\{f(a_t) - f(a)\} + \{f(a_t') - f(a)\}\right)\right] \\
&= l_0 Reg_T^{FO}.
\end{aligned}
$$

The first inequality of (11) can be proven in a similar manner. ∎

## Footnotes

[1] Although the regret in (2) appears superficially different from that in Yue and Joachims (2009), two regrets can be shown to coincide with each other under Assumptions 1-3 in Subsection 1.2.

[2] The optimal order changes depending on the model parameter $\kappa \geq 1$ of the pairwise comparison oracle in Jamieson et al. (2012).