[Reviews · NeurIPS 2017]

Reviewer 1



The paper analyzes the dueling bandit problem when the action space is a convex and compact subset of the d-dimensional Euclidean space. Moreover, the noisy comparison feedback for the choice (a,a') is such that P(a > a') can be written as sigma(f(a') - f(a)) where f is a strongly convex and smooth function and sigma is a sigmoidal link function. The player's goal is to minimize a notion of regret which, under the above assumptions, is equivalent to previously proposed notions. The proposed algorithm uses ideas from convex bandit optimization with self-concordant regularization. In essence, the dueling bandit regret is shown to be upper bounded by an expression that is similar to a convex bandit regret with a loss function P_t that is locally strongly convex and smooth. At each time step, the player uses (a_t,a_t') where a_t is determined through stochastic gradient descent and a_t' is used to define a stochastic gradient estimate. Additional (simpler) results relate the dueling bandits regret bounds to regret bounds in stochastic convex optimization with pairwise comparison feedback. Via the adaptation of known lower bounds for stochastic optimization, the regrets are shown to be optimal ignoring log factors. The paper builds on previous results by [9] on convex bandit optimization which used similar strongly convex / smooth assumptions. However, the adaptation to dueling bandits is not trivial. Overall, the paper looks technically strong with interesting results. I am a bit disappointed by the general lack of intuition in explaining the approach and the algorithm. Also, the related work section does not compare with the rates obtained in previous works, especially [2] which also used convex action sets, even though the assumptions are not identical.

Reviewer 2



The submitted paper presents a stochastic mirror descent algorithm for the dueling bandits problem defined in Yue & Joachims 2009. The paper gives a regret bound based on a regret definition established in the dueling bandits framework (where two nearly optimal arms have to be selected in order to minimize regret), an connects this result to convergence rates with respect to the optimality gap in convex optimization. The latter result is the most interesting contribution of the paper. While the paper is self-contained and the reviewer could not find an error in the proofs (not being an expert in bandit learning), the paper could be improved considerably by putting it into the greater context of stochastic zero-order convex optimization. The gradient estimation approach given in Section 3.2 is reminiscent of work by Duchi et al. (2015). Optimal Rates for Zero-Order Convex Optimization: The Power of Two Function Evaluations. The convergence analysis given in the cited paper can also be interpreted as an expected regret bound for online bandit learning. How do the convergence rates compare? Earlier work on zero-order optimization by Ghadimi & Lan or Nesterov & Spokoiny is cited in this paper. Furthermore, it would be nice to get a clearer picture how the presented work is different from related work in online bandit learning. Wouldn't the analysis of Yue & Joachims 2009 be easily extended to the setting of the paper by adding an appropriate proximal operator to their algorithm? Where does the assumption of linearity restrict Zhang et al.'s analysis to apply to the presented convex optimization framework? Overall recommendation: The paper is certainly theoretically solid - a bit more explanation would not hurt and possible lead to wider reception of the presented work.